# An Underwater Image Enhancement Method for a Preprocessing Framework Based on Generative Adversarial Network

**DOI:** 10.3390/s23135774

**Published:** 2023-06-21

**Authors:** Xiao Jiang, Haibin Yu, Yaxin Zhang, Mian Pan, Zhu Li, Jingbiao Liu, Shuaishuai Lv

**Affiliations:** 1College of Electronics and Information, Hangzhou Dianzi University, Hangzhou 310018, China; 2143040030@hdu.edu.cn (X.J.); shoreyhb@hdu.edu.cn (H.Y.); bruce_zyx@hdu.edu.cn (Y.Z.); ai@hdu.edu.cn (M.P.); lz1126@hdu.edu.cn (Z.L.); 2Ningbo Institute of Oceanography, Ningbo 315832, China; 3Ocean Technology and Equipment Research Center, Hangzhou Dianzi University, Hangzhou 310018, China; ab@hdu.edu.cn

**Keywords:** underwater image enhancement, convolutional neural network (CNN), generative adversarial networks (GANs), feature extraction, cross-stage fusion

## Abstract

This paper presents an efficient underwater image enhancement method, named ECO-GAN, to address the challenges of color distortion, low contrast, and motion blur in underwater robot photography. The proposed method is built upon a preprocessing framework using a generative adversarial network. ECO-GAN incorporates a convolutional neural network that specifically targets three underwater issues: motion blur, low brightness, and color deviation. To optimize computation and inference speed, an encoder is employed to extract features, whereas different enhancement tasks are handled by dedicated decoders. Moreover, ECO-GAN employs cross-stage fusion modules between the decoders to strengthen the connection and enhance the quality of output images. The model is trained using supervised learning with paired datasets, enabling blind image enhancement without additional physical knowledge or prior information. Experimental results demonstrate that ECO-GAN effectively achieves denoising, deblurring, and color deviation removal simultaneously. Compared with methods relying on individual modules or simple combinations of multiple modules, our proposed method achieves superior underwater image enhancement and offers the flexibility for expansion into multiple underwater image enhancement functions.

## 1. Introduction

Underwater vision has emerged as a vital tool for exploring the marine environment, offering non-invasive access and rich information content. High-quality underwater images play a crucial role in providing valuable information for underwater robots engaged in various missions, such as underwater exploration, marine archaeology, underwater rescue, and underwater imaging. However, underwater images are prone to color distortion, low contrast, and motion blur due to the absorption and scattering of light [1,2,3,4,5]. Consequently, enhancing the quality of underwater optical imaging has become a significant research focus in the field of ocean engineering and computer vision [6,7,8,9,10].

Over the past few decades, researchers and experts have devoted substantial efforts to underwater image enhancement. Various methods can be categorized as physical model-based [1,2,3,4,5], non-physical model-based [6,7,8,9], and data-driven approaches [10,11,12,13,14,15,16,17,18,19,20,21,22]. Physical model-based methods treat underwater image enhancement as an inverse problem and rely on estimating the parameters of the image formation model to obtain clear underwater images. One of the most widely used methods is the dark channel prior (DCP) model, including underwater dark channel prior (UDCP) [1] and generalized dark channel prior (GDCP) [2]. Another approach considers the optical characteristics of underwater images, such as forecasting scene transmission using the attenuation differences of RGB color channels [3], underwater image enhancement based on minimum information loss and histogram prior distribution considering the relationship between background color and optical properties of the medium [4,5], and underwater image color correction methods [6]. However, physical model-based methods suffer from two main drawbacks. First, they often require prior knowledge due to the complex underwater environment, limiting their effectiveness. Second, these methods rely on simplified image formation models that assume uniform decay coefficients across color channels, resulting in unstable enhancements and suboptimal visual effects.

In contrast to physical model-based methods, non-physical model-based approaches directly manipulate pixel values to enhance underwater images. These methods include white balance [7], grayscale world theory [8], and histogram equalization [9]. For instance, a white balance method presented in [7] enhances underwater images through gamma correction and sharpening, whereas a variational Retinex method proposed in [8] involves color correction, layer decomposition, and late enhancement. Furthermore, a two-step approach combining color correction and contrast enhancement was introduced in [9]. However, these methods often overlook the unique optical properties of underwater images, leading to color biases and oversaturation or undersaturation in different regions. They also fail to fully consider the degradation mechanisms, resulting in red artifacts and potential introduction of pseudo-targets.

As deep learning methods are widely used in image processing and computer vision, enhanced methods for underwater images have been developed rapidly. Chang et al. [23] systematically discussed the application of generative adversarial networks (GANs) in scene graphs. Zhang et al. [24] highlighted the transferability aspect of the proposed method. In zero-shot learning [25] and unsupervised methods [26], GANs demonstrate their flexibility in application. By designing end-to-end networks, complex underwater image degradation models can be avoided. Typical underwater image enhancement methods include GAN-based and convolutional neural network-based (CNN-based) models. Fabbri et al. [10] employed conditional GAN (cGAN) to tackle underwater image enhancement as an image-to-image conversion problem. Building upon this work, Yu et al. [11] incorporated perceived loss into the cGAN framework for underwater image color correction. Subsequent studies explored multi-scale dense GANs for underwater image enhancement [12,13]. Liu et al. proposed a deep multiscale feature fusion network for underwater image color correction. However, GAN-based underwater image enhancement methods heavily rely on aligned underwater image pairs, which are often challenging to obtain. Additionally, most of these methods primarily focus on color correction and overlook overall detail enhancement, leading to suboptimal visual effects. To address these issues, unsupervised underwater image enhancement methods have been proposed [12,15,16]. Li et al. [15] introduced WaterGAN, a GAN-based algorithm that generates realistic underwater images from aerial images and their corresponding depth maps, providing unsupervised color correction. Islam et al. [16] proposed a rapid underwater image enhancement method for real-time preprocessing in an autonomous pipeline using a CycleGAN-based visual-guided underwater robot. Furthermore, Hong et al. [20] proposed a weakly-supervised underwater image enhancement (WSUIE) method to reduce reliance on raw/enhanced underwater image alignment. However, the practical application of these algorithms is significantly limited due to the difficulty in obtaining corresponding depth maps.

In CNN-based underwater image enhancement, Anwar et al. [18] introduced UWCNN, an end-to-end model trained on a synthetic underwater image dataset. Sun et al. [19] proposed a deep model for underwater image enhancement using an encoder–decoder framework with skip connections to preserve low-level features and expedite training. Wu et al. [21] developed an underwater image CNN (UWCNN-SD) based on structure decomposition to address color distortion, blurred details, and low contrast in underwater images. Li et al. [22] proposed an adaptive algorithm utilizing random wired neural networks (RWNN) and co-evolution (SE) to perform color adjustment, contrast improvement, luminance enhancement, and detail enhancement with edge retention techniques. However, despite the powerful feature extraction capabilities of CNNs, most CNN-based underwater image enhancement algorithms operate in the RGB color space. Although the RGB color space can address scattering issues and improve color deviation, it fails to directly capture critical parameters related to quality degradation, such as low contrast, saturation, and luminance.

Given the complex light propagation characteristics in the ocean, leading to severe color distortion, low contrast, and motion blur in underwater robot photography, this paper proposes an efficient image preprocessing framework called ECO-GAN based on a generative adversarial network. The framework enables blind enhancement of individual underwater images. ECO-GAN incorporates a CNN to tackle three underwater problems: dynamic blur, low brightness, and color bias. Firstly, we utilize the U-Net structure to sample the input underwater image and extract features of different scales through the encoder. Subsequently, by analyzing the multi-scale features and under the constraint of the objective function, we alleviate the quality issues while upsampling and recovering the image to its original size, thereby achieving blind image denoising (removing unknown noise from noisy images). The contributions of this article are as follows:

(1) We propose a generative adversarial network capable of efficiently addressing various image enhancement tasks for underwater images with multiple defects.

(2) The designed image enhancement model enables simultaneous enhancement of three distinct image enhancement tasks. By employing feature extraction through repeated encoders and dedicated decoders for different enhancement tasks, computational complexity is reduced and inference speed is improved.

(3) ECO-GAN incorporates cross-stage fusion modules between decoders for multiple enhancement tasks, thereby enhancing the interconnection between decoders and improving the quality of output images.

## 2. Underwater Image Enhancement Network

Figure 1 presents the block diagram of the ECO-GAN structure for underwater image enhancement. ECO-GAN consists of a generator and three discriminators. During the training stage, original underwater images are input to the generator to learn the corresponding multiscale residual map, recovering three potentially clear images at different upsampling stages. These results are then passed to the corresponding discriminator, which independently assesses their authenticity at different scales. In the inference stage, the generator enables end-to-end underwater image enhancement for input underwater images. Now, we will delve into the details of the generator and the discriminator.

### 2.1. Generator

The generator structure of ECO-GAN, as depicted in Figure 2, comprises three parts: downsampled feature extraction, upsampled image recovery, and cross-stage fusion. The main functions of the generator include luminance enhancement, blur removal, and white balance.

To reduce computational complexity, we reused the backbone feature extraction network and extracted features only once. This design was motivated by Zhang et al. [27], who implemented three separate functions and combined them in series. However, repeatedly extracting features three times can be computationally expensive. In our approach, we employed a classical U-Net structure and proposed a cross-stage fusion module to establish connections between the three branches. The U-Net structure resembles the letter “U” and enables efficient feature extraction and image recovery. The cross-stage fusion module is to combine the feature extracted by current-stage network and last-stage network.

As shown in Figure 2, our model is designed to realize three tasks and reuse backbone for reducing calculations. To recover the raw input image $I_{in}$ to the final image $I_{f}$, we utilize a function F(·) to extract features at different scales and a function G(·) to recover them.
(1)$$I_{f}=G(F(I_{in}))$$

For the three specific tasks, we introduce three different functions, namely, $G_{1}$, $G_{2}$, and $G_{3}$, to recover extracted features. Our target tasks are specified by paired datasets.
(2)$$output1=G_{1}(F(I_{in}))$$
(3)$$output2=G_{2}(F(I_{in}))$$
(4)$$output3=G_{3}(F(I_{in}))$$
where output1, output2, and output3 are outputs of $G_{1}$, $G_{2}$, and $G_{3}$, respectively.

To integrate these tasks, we need $G_{t}$, which utilizes the features extracted by F and recovered by $G_{1}$, $G_{2}$, and $G_{3}$ at different scales. We can express the estimate of the final enhanced image as follows:$${\hat{I}}_{f}=G_{t}\left(F\left(I_{in}\right),G_{1}\left(F\left(I_{in}\right)\right),G_{2}\left(F\left(I_{in}\right)\right),G_{3}\left(F\left(I_{in}\right)\right)\right)$$
(5)$$={G_{t}}^{\prime }\left(F\left(I_{in}\right)\right)$$
where ${\hat{I}}_{f}$ is the estimate of $I_{f}$; $G_{t}$ is a function that can utilize features extracted by F and recovered by $G_{1}$, $G_{2}$, and $G_{3}$ at different scales; and ${G_{t}}^{\prime }$ is an alternate form of $G_{t}$.

To optimize the model and reduce computational costs, we leverage a backbone feature extraction network to extract features at different scales in the downsampling stage. The backbone feature extraction network adopts a modular design, incorporating convolutional layers, batch normalization layers, and activation layers. The outputs of each downsampling module are connected to the corresponding module in the upsampling image recovery stage through skip connections. This approach effectively addresses the issue of gradient disappearance. In the upsampling stage of image recovery, we employ three upsampled image recovery branches that correspond to the three image enhancement tasks: luminance enhancement, blur removal, and white balance. The image recovery branch also adopts a modular design, where each module consists of deconvolution, batch normalization layers, and activation layers.

Both the backbone feature extraction network and the image recovery branches follow a U-Net structure. The downsampling module utilizes skip connections to propagate extracted features to the subsequent upsampling recovery module, mitigating gradient disappearance and guiding the direction of image repair. This is crucial because the upsampling module outputs variations in the image rather than the image itself.

In the upsampling stage, an input image undergoes three separate image repair branches to achieve blur removal, low-light enhancement, and color bias correction. To improve the quality of image recovery, a cross-stage fusion module is incorporated to integrate the recovery features from different enhancement tasks. The cross-stage fusion module concatenates input features from different stages, adding two channels of ‘ch’ along the channel dimension to aid the recovery task at the current stage. This results in a channel number of 2 × ch while maintaining the original width and height of the image. The number of channels is subsequently reduced to ‘ch’ through convolutional layers, generating features that retain the original input feature size and contain multistage information. Additionally, the cross-stage fusion module stacks the features of the previous stage and the current stage. By introducing the features of the previous stage, it assists with the recovery task of the current stage and further integrates the features through convolution, maintaining the feature dimension. As a single model, the entire image enhancement model can accomplish various enhancement tasks, striking a balance between computational efficiency and performance.

### 2.2. Discriminator

The ECO-GAN discriminator distinguishes image blocks of size 70 × 70 to obtain local-scale discrimination results and combines them with a standard global discriminator to assess the entire image. This approach yields a two-scale least squares discriminator. In this study, we adopt a two-scale least squares discriminator to determine the authenticity of the image. The two-scale representation assesses the authenticity of the input image from both a global semantic level and local detail level. This guides the generator to generate real images at both scales, facilitating the production of realistic image details that improve visual results. Since the ECO-GAN handles three image enhancement tasks, we utilize three multilayer perceptrons as discriminant networks to evaluate the output of each generator separately. The structure of the discriminator is illustrated in Figure 3, where ‘output *i*’ (*i* = 1, 2, 3) refers to the three images generated by the generator.

### 2.3. Loss Function

#### 2.3.1. Loss Function of the Generator

The loss function for the ECO-GAN generator is complex, encompassing multiple components. It includes blur removal loss, low-light enhancement loss, and color bias correction loss. From the neural network perspective, it comprises the generator loss $L_{G}$ and the discriminator loss $L_{D}$. The generator loss further consists of pixel detail loss, abstract content loss, and adversarial loss.
(6)$$L_{G}=L_{db-G}+L_{lt-G}+L_{wb-G}$$
(7)$$L_{D}=E\left(D\left(img^{\prime }\right)\right)-E\left(D\left(img^{gt}\right)\right)+gp$$
(8)$$gp=\lambda E\left[{\left({\left|\nabla_{{img}^{gt}}D\left(img^{gt}\right)\right|}_{2}-1\right)}^{2}\right]$$
where the corresponding discriminator of each branch adopts the same loss function, and $L_{D}$, $img,\;img^{\prime },\;and\;img^{gt}$ represent the input original image, the generated image of the generator output, and the label truth-value image, respectively. A gradient penalty term proposed by Gulrajani et al., namely, Formula (3), is referred to here, and it is applicable to multiple generator structures and requires little adjustment of hyperparameters, i.e., λ=10 in this experiment.

The loss function of the generator for the blur removal branch is as follows:(9)$$L_{db-G}=\sum_{pix}^{pixels}|{\left.{img}^{\prime }-img^{gt}\right|}^{2}+L_{X-db}+Loss_{db}^{RaLSGAN}\left(img,img^{gt}\right)$$

The loss function of the generator for the low-light enhancement branch is as follows:(10)$$L_{lt-G}=\sum_{pix}^{pixels}|{\left.{img}^{\prime }-img^{gt}\right|}^{2}+L_{X-lt}+Loss_{lt}^{RaLSGAN}\left(img,img^{gt}\right)$$

The loss function of the generator for the color bias correction branch is as follows:(11)$$L_{wb-G}=\sum_{pix}^{pixels}|{\left.{img}^{\prime }-img^{gt}\right|}^{1}+L_{X-wb}+Loss_{wb}^{RaLSGAN}\left(img,img^{gt}\right)$$
(12)$$L_{X}=\frac{1}{W_{i,j}H_{i,j}}\sum_{x=1}^{W_{i,j}}\sum_{y=1}^{H_{i,j}}{\left(M_{i,j}\left({img}^{\prime }\right)-M_{i,j}\left(img^{gt}\right)\right)}^{2}$$
where $M_{i,j}$ represents the feature map between the convolutional layer *i* and convolutional layer *j* of the convolutional neural network VGG19, and $W_{i,j}H_{i,j}$ is the height and width between the convolutional layer *i* and convolutional layer *j*.

#### 2.3.2. The Loss Function of the Discriminator

To make the discriminator training faster and the network more stable, we use the RaGAN-LS loss function proposed by Tetiana Martyniuk et al. in [28]:$$Loss^{RaLSGAN}\left(x,z\right)=E_{x\sim P_{data}\left(x\right)}\left[{\left(D\left(x\right)-E_{z\sim P_{z}\left(z\right)}\left[D\left(G\left(z\right)\right)\right]-1\right)}^{2}\right]+$$
(13)$$E_{z\sim p_{z}\left(z\right)}\left[{\left(D\left(G\left(z\right)\right)-E_{x\sim P_{data}\left(x\right)}\left[D\left(x\right)\right]+1\right)}^{2}\right]$$
where *x* represents the input data, which follows the distribution $P_{data}(x)$ and represents the data generated by the generator, $\tilde{x}=G\left(z\right)$; *z* represents a sample from the noise distribution, which follows the distribution $P_{z}\left(z\right)$; *G* and *D* represent the generator model and the discriminator model, respectively; and E indicates the expectation.

## 3. Experiment

### 3.1. Evaluation Indicator

In this paper, we utilized PSNR as an evaluation index to measure the enhancement effect of ECO-GAN. The width and height of a given image are *h* and *w*, respectively. The enhanced image is recorded as $I_{c}$, whereas the original noise image is recorded as $I_{n}$. The mean square error (MSE) of the enhanced and original images is defined as $MSE=\frac{1}{hw}\sum_{i=0}^{h-1}\sum_{j=0}^{w-1}{\left[I_{c}\left(i,j\right)-I_{n}\left(i,j\right)\right]}^{2}$, and the PSRN (dB) is defined as:(14)$$PSNR=10\cdot \log_{10}\left(\frac{MAX_{I}}{MSE}\right)$$
where $MAX_{I}$ is the maximum pixel value of the image. If each pixel is represented by a B-bit binary number, then $MAX_{I}=2^{B}-1$. In this paper, if each pixel is represented by an 8-bit binary number, then the $MAX_{I}$ is 255. In this paper, we accumulated PSNR of the three channels of RGB.

In addition, we used structural similarity SSIM to measure the luminance, contrast, and structure (structure) between samples *x* and *y*.
(15)$$l\left(x,y\right)=\frac{2\mu_{x}\mu_{y}+c_{1}}{\mu_{x}^{2}+\mu_{y}^{2}+c_{1}}$$
(16)$$c\left(x,y\right)=\frac{2\sigma_{x}\sigma_{y}+c_{2}}{\sigma_{x}^{2}+\sigma_{y}^{2}+c_{2}}$$
(17)$$s\left(x,y\right)=\frac{\sigma_{xy}+c_{3}}{\sigma_{x}\sigma_{y}+c_{3}}$$
where $\mu_{x}$ and $\mu_{y}$ are the mean of *x* and *y*, respectively; $\sigma_{x}^{2}$ and $\sigma_{y}^{2}$ are the variance of *x* and *y*, respectively; $\sigma_{xy}$ is the covariance of *x* and *y*; and $c_{1}={\left(k_{1}MAX_{I}\right)}^{2}$ and $c_{2}={\left(k_{2}MAX_{I}\right)}^{2}$ are the two constants. Take $c_{3}=c_{2}/2$ and avoid dividing by zero. $MAX_{I}$ represents the maximum value of pixels in the B-bit image, which is 255 in this paper.

Take $k_{1}=0.01,\;k_{2}=0.03$ as the default value, then
(18)$$SSIM\left(x,y\right)=\left[l{\left(x,y\right)}^{\alpha }\cdot c{\left(x,y\right)}^{\beta }\cdot s{\left(x,y\right)}^{\gamma }\right]$$

Set α, β, γ to 1, then
(19)$$SSIM\left(x,y\right)=\frac{\left(2\mu_{x}\mu_{y}+c_{1}\right)\left(2\sigma_{xy}+c_{2}\right)}{\left(\mu_{x}^{2}+\mu_{y}^{2}+c_{1}\right)\left(\sigma_{x}^{2}+\sigma_{y}^{2}+c_{2}\right)}$$

### 3.2. Experimental Results

The data used in this experiment were provided by the Underwater Robot Picking Competition (URPC) organized by the National Natural Science Foundation of China. The dataset used in this paper was URPC2019, which consists of images captured by an underwater robot with a camera and primarily exhibits quality issues such as blur, low light, and color distortion. We used Adobe’s LightRoom Classic software (version 10.0) for batch processing of the images. The dataset included 5543 images, which were divided into a training set and a test set in a 7:3 proportion. The training set comprised 3880 deblurred truth-value images, 3880 enhanced truth-value images, and 3880 dedistorted truth-value images. The test set consisted of 1663 deblurred truth-value images, 1663 enhanced truth-value images, and 1663 dedistorted truth-value images. The proposed ECO-GAN and the comparison model were executed on a single NVIDIA GeForce RTX 2080Ti video card.

To validate the effectiveness of the luminance enhancement, blur removal, and color bias correction modules in the generator, separate experiments were conducted on each module, and the three modules were also tested in combination to assess their individual and combined efficacy. The experimental results are presented in Table 1, which demonstrates that the PSNR and SSIM indices for image enhancement using the proposed ECO-GAN method outperformed both individual and combined enhancement modules. In addition, we used UIQM and UCIQE, which are non-reference evaluation metrics. In Table 1, GT serves as the ground truth for calculating reference evaluation indicators. Light, Sharp, and WB represent modules for low-light enhancement, blur removal, and color bias correction, respectively. “Light + Sharp” represents a combination of a low-light enhancement module and a blur removal module. “Light + WB” represents a combination of a low-light enhancement module and a bias correction module. “Sharp + WB” represents a combination of a blur removal module and a bias correction module. “Light + Sharp + WB” represents a combination of a low-light enhancement module, a blur removal module, and a color bias correction module. The combination method among three modules is in series. In order to verify the superiority of the proposed ECO-GAN, we compared ECO-GAN with the method proposed by Zhang et al. in [27] using a similar three-module serial structure. Due to the enhanced intercorrelation between multiple modules achieved through repeated feature extraction in the U-Net, the three enhancement modules collectively performed better than their independent counterparts, thereby facilitating flexible expansion of image enhancement capabilities. Moreover, Zhang et al. [24] highlighted the transferability aspect, which aids in accelerating model stability. We observed that ECO-GAN loaded with a pretrained model achieved faster stability compared with ECO-GAN without pretrained weights.

The underwater image enhancement results obtained by different methods are presented in Figure 4. The raw image (Figure 4a) represents the original underwater image and serves as the input of the algorithm. GT (Figure 4b) serves as the ground truth for calculating reference evaluation indicators. Figure 4c–e show the enhanced results of GT after being processed by three modules, namely, “Light,” “Sharp,” and “WB,” respectively. Meanwhile, they were used to calculate the loss between corresponding image enhancement results. Figure 4f–k show the results of the single module and pairwise combinations of three modules on the raw image. Figure 4l–n show the results of the serial combination of three modules, the method proposed in [27], and the proposed ECO-GAN applied to the raw image, respectively. Due to the low quality of images in harsh underwater scenes, the non-reference values of both truth and result images are generally low. For reference indicators, the results demonstrate that ECO-GAN exhibited excellent performance in underwater image enhancement tasks.

## 4. Discussion

In this paper, we propose an underwater image enhancement method, ECO-GAN, based on an image preprocessing framework for GANs. Our method addresses three key challenges in underwater images: dynamic blur, low-light conditions, and color bias. The goal is to achieve blind enhancement of underwater images with multiple defects. To reduce computational effort and enhance inference speed, we employ the U-Net network structure and design decoders for different enhancement tasks by reusing the encoded features. By analyzing multi-scale features, we aim to alleviate image-quality issues. Additionally, a cross-stage fusion module is employed between the decoders for multiple enhancement tasks, strengthening their connection and enabling the recovery of details and improved image contrast. During the training process, the images are blindly enhanced using supervised training on paired datasets, eliminating the need for additional physical knowledge or prior information.

The proposed method significantly enhances the visualization of underwater images and facilitates the implementation of vision-based underwater tasks such as segmentation and tracking. In future work, we plan to explore the application of our method in related areas such as image defogging and super-resolution reconstruction to evaluate its generality. Additionally, we aim to apply the efficient model to unsupervised learning. These avenues remain as potential directions for future research.

## Figures and Tables

**Figure 1 sensors-23-05774-f001:**
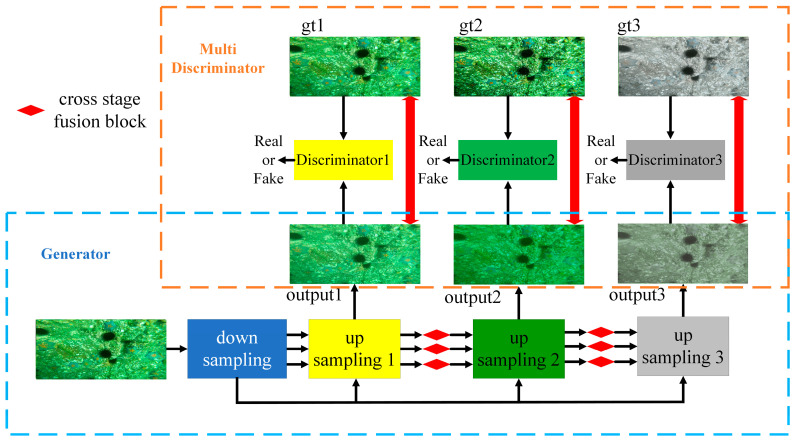
Overview of the proposed ECO-GAN for underwater image enhancement.

**Figure 2 sensors-23-05774-f002:**
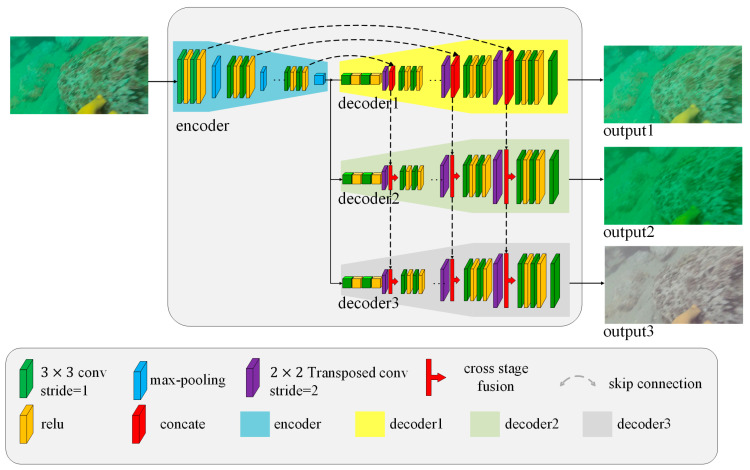
Structure of ECO-GAN generator.

**Figure 3 sensors-23-05774-f003:**
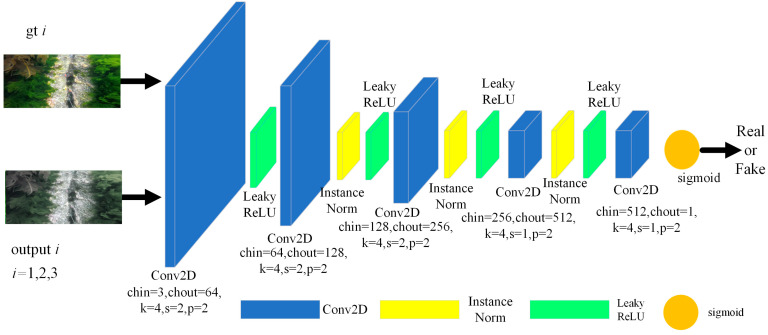
ECO-GAN discriminator.

**Figure 4 sensors-23-05774-f004:**
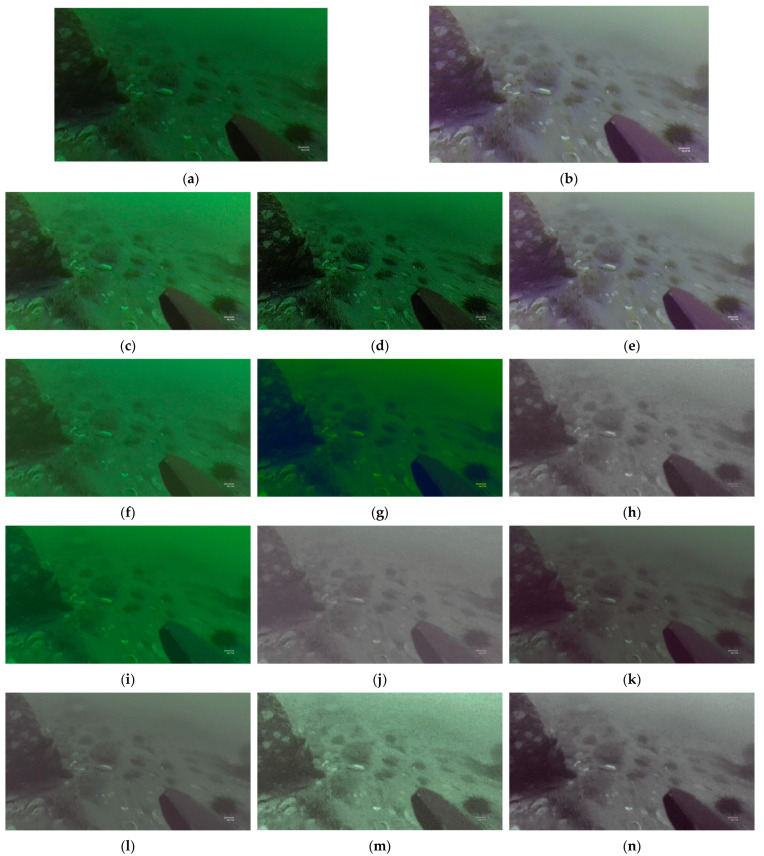
Underwater image enhancement results using different methods. (**a**) Raw; (**b**) GT; (**c**) Light on GT; (**d**) Sharp on GT; (**e**) WB on GT; (**f**) Light on raw; (**g**) Sharp on raw; (**h**) WB on raw; (**i**) “Light + Sharp” on raw; (**j**) “Light + WB” on raw; (**k**) “Sharp + WB” on raw; (**l**) “Light + Sharp + WB” on raw; (**m**) Method proposed in [27] on raw; (**n**) Proposed ECO-GAN on raw.

**Table 1 sensors-23-05774-t001:** Comparison of experimental results.

Method			PSNR	SSIM	UIQM	UCIQE
GT			/	1.0	0.2337	0.2524
Light			16.19	0.9075	0.1404	0.2166
	Sharp		11.48	0.7224	0.1467	0.2094
		WB	31.40	0.9613	0.1744	0.2191
Light	Sharp		11.95	0.7609	0.1523	0.2125
Light		WB	24.56	0.9587	0.1867	0.2214
	Sharp	WB	21.79	0.9487	0.1899	0.2170
Light	Sharp	WB	24.12	0.9521	0.1985	0.2263
Method proposed by Zhang et al. in [27]			17.29	0.7648	0.1556	0.1678
ECO-GAN			32.71	0.9783	0.1970	0.2016

## Data Availability

Not applicable.

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
