# Peer review of "An Underwater Image Enhancement Method for a Preprocessing Framework Based on Generative Adversarial Network"

_sensors, 2023, doi:10.3390/s23135774_

Round 1
Reviewer 1 Report
1. In Figure 1, the images obtained at different stages look similar, e.g. output1, output2, output3, gt1 gt2, gt3;
2. The content of lines 155-159 is not relevant to the research of the manuscript, which should be the instructions for the paper template?
3. In Figure 2, the meaning of each color block should be given, so that it can be more easily understood by the reader.
4. In Section 2.1, "During the upsampling stage of image recovery, the design uses three upsampled image repair branches corresponding to three image enhancement tasks - luminance enhancement, blur removal and white balance." Why do the three upsampled branches mentioned in this sentence correspond to image luminance enhancement, blur removal and white balance?
5. In Section 2.1, the authors only describe the process of implementing the ECO-GAN generator and do not elaborate on the relevant theoretical rationale for doing so.
6. In section 2.2, the statement "ECO-GAN discriminator of eco-gan distinguishes the image block of size 7070" is unclear.
7. In Section 2.3.1, the corresponding reference for the RaGAN-LS loss function is not given.
8. In section 3.1, "PSRN" should be "PSNR". Also, should "12" in "the enhanced image is recorded as: 12" be Ic?
9. The dataset used in the paper is from URPC, which year of the competition dataset is it? This should be specified.
10. The method in this paper is not compared with other recent algorithms, which cannot prove the superiority of the algorithm.
11. The evaluation metrics used in the paper are PSNR and SSIM, and it is suggested to add the commonly used non-reference evaluation metrics, UIQM and UCIQE.
English writing needs to be improved.
Author Response
Dear reviewers,
We thank you for their careful review of our paper and the many valuable comments and suggestions they you provided. We believe that these have enabled us to greatly improve the quality of our paper. We have addressed all of the reviewer comments, and we have provided our point-by-point responses to them in three separate files. Please see the attachment file for detail.

Reviewer 2 Report
The paper presents an underwater image enhancement method using a GAN-based preprocessing framework. The proposed ECO-GAN method addresses several challenges encountered in underwater imaging, including color distortion, low contrast, and motion blur. The paper provides a detailed description of the framework, architecture, and training process. Overall, the paper contributes to the field of underwater image processing and provides a comprehensive solution to enhance the quality of underwater images.
Specific Comments:
-
The abstract provides a clear overview of the paper. However, it would be helpful to include a sentence or two summarizing the key findings or contributions of the proposed method. This would give readers a better understanding of the significance of the work right from the beginning.
-
The methodology section provides a detailed description of the proposed ECO-GAN framework. However, some of the technical terms and concepts introduced are not adequately explained. It would be beneficial to provide clearer explanations or definitions for terms such as "U-Net structure," "cross-stage fusion modules," and "blind image denoising" to ensure readers can follow the methodology effectively.
-
The authors mention the use of a "convolutional neural network" to solve underwater problems, but they do not provide specific details about this CNN. It is crucial to provide information regarding the architecture, such as the number of layers, types of layers (e.g., convolutional, pooling), and any unique design choices made.
-
The evaluation and experimental results section presents a comparison between the proposed method and other approaches. While the results indicate the effectiveness of the proposed method, it would be beneficial to include more comprehensive comparisons with state-of-the-art methods to demonstrate the superiority of ECO-GAN more convincingly.
-
The paper should cite "A Comprehensive Survey of Scene Graphs: Generation and Application" to provide a broader context for the use of generative models in computer vision tasks. This survey paper covers the generation and application of scene graphs, which are relevant to the proposed method's image enhancement process.
-
The paper should cite "TN-ZSTAD: Transferable Network for Zero-Shot Temporal Activity Detection" as it introduces a transferable network for zero-shot temporal activity detection. This reference is relevant because it highlights the transferability aspect of the proposed method, which aims to enhance underwater images without relying on additional physical knowledge or prior information.
-
The paper should cite "Video Pivoting Unsupervised Multi-Modal Machine Translation" as it explores unsupervised multi-modal machine translation in the context of videos. This reference is relevant because it demonstrates the potential of unsupervised methods in handling multi-modal data, which can be beneficial in underwater image enhancement.
-
The paper should cite "ZeroNAS: Differentiable Generative Adversarial Networks Search for Zero-Shot Learning" as it presents a differentiable approach for searching generative adversarial networks (GANs) in zero-shot learning. This reference is relevant because it showcases a similar research direction of using GANs in zero-shot scenarios, which aligns with the proposed method's blind image denoising aspect.
N/A
Author Response

(The authors gave the same response as above.)

Round 2
Reviewer 1 Report
In Figure 3, gt i and output i look similar. Please distinguish them.
Author Response
We apologize for using a pair of images with low discrimination. We have replaced the original image pair with a pair of clearly distinguishable images in Figure 3 in the revised manuscript (Line 214). Please see the revised version for detail.